# Learning where to intervene with a differentiable top-K operator: Towards data-driven strategies to prevent fatal opioid overdoses

Kyle Heuton [1]   Shikhar Shrestha [2]   Thomas J. Stopka [2]   Michael C. Hughes [1]

## Abstract

To mitigate the ongoing opioid overdose crisis, public health organizations need to decide how to prioritize targeted interventions in the most effective manner, given many candidate locations but a limited budget. We consider learning from historical opioid overdose events to predict where to intervene among many candidate spatial regions. Recent work has suggested performance metrics that grade models by how well they recommend a top-K set of regions, computing in hindsight the fraction of events in the actual top-K regions that are covered by the recommendation. We show how to directly optimize such metrics, using advances in perturbed optimizers that allow end-to-end gradient-based training. Experiments suggest that on real opioid-related overdose events from 1620 census tracts in Massachusetts, our end-to-end neural approach selects 100 tracts for intervention better than purpose-built statistical models and tough-to-beat historical baselines.

## 1. Introduction

The opioid overdose epidemic in the United States has resulted in approximately 400,000 deaths over the past two decades, with more than 100,000 fatal overdoses alone from April 2020 to April 2021, a 28.5% increase over the prior 12-month period (U.S. CDC, 2022). The severity of this overdose epidemic varies widely across space and time (Li et al., 2019; Shover et al., 2020), influenced by multi-level factors from drug supply to local socio-demographics to access to prevention interventions and substance use treatment. There is a need for prediction models that can overcome this heterogeneity to make accurate predictions of near-term future overdose events at fine spatiotemporal resolutions, so that public health agencies can decide how to allocate limited resources towards the goal of harm reduction. In this work, we investigate the possibility of learning from past event counts over a spatiotemporal grid with $S$ regions and $T$ timesteps to recommend regions that should implement evidence-based interventions in the near future.

Choosing sensible performance metrics is critical to understanding which forecasting models have real-world utility. Because fatal overdoses are a relatively rare outcome, the observed event count time series exhibits high sparsity (Heuton et al., 2022) and thus traditional evaluation metrics such as Root Mean Squared Error (RMSE) may not be the most suitable. In a recent plan for assessing spatiotemporal forecasts of opioid overdose events, Marshall et al. (2022) suggest a new metric: first ask the model to recommend $K$ regions (where $K \ll S$) thought to be the highest-risk in near-future, then in hindsight record the proportion of all fatal overdoses events captured or *reached* in this subset. Heuton et al. (2022) call this metric BPR, short for percentage of best-possible reach. Simply put, this metric is *intervention-aware*: public health agencies combating the opioid overdose epidemic have limited budgets, and may only be able to deploy some high-cost evidence-based interventions in targeted higher-risk areas. BPR rewards models that correctly identify the highest need areas. The size of the subset $K$ can be selected to match real-world budgets.

Our team has found that several well-known machine learning (ML) approaches are underwhelming in terms of this preferred BPR performance metric. Simple baselines that "predict" using averages of historical mortality score about as well as do more sophisticated models (see Tab. 1), including ensembles of trees and our previous work on Gaussian Processes (Heuton et al., 2022). This difficult forecasting problem clearly presents challenges to current ML methods. New approaches are needed.

We suggest that directly optimizing the quantity of interest, BPR, will lead to more useful models. Taking inspiration from work on *direct loss minimization* (Wei et al., 2021; Lacoste–Julien et al., 2011), we pursue in this paper an

---

[1]Department of Computer Science, Tufts University, Medford, MA, USA [2] Dept. of Public Health and Community Medicine, Tufts University School of Medicine, Boston, MA, USA. Correspondence to: Michael C. Hughes <michael.hughes@tufts.edu>.

*Workshop on Interpretable ML in Healthcare at International Conference on Machine Learning (ICML)*, Honolulu, Hawaii, USA. 2023. Copyright 2023 by the author(s).

Code URL: github.com/tufts-ml/differentiable-top-k

approach that directly trains models to score all $S$ spatial regions for near-future risk, then uses these scores to select a top-K set of regions that will deliver high BPR. Gradient-based training of such models is challenging because of the discrete ranking involved: slight changes to parameters will almost always leave rankings unchanged, making acquiring signal for how to improve parameters difficult. We build upon work by Cordonnier et al. (2021), who show how to select a top-K set of patches from a high-resolution image for downstream supervised tasks. Their approach (and ours) uses a differentiable top-K operator based on perturbed optimizers (Berthet et al., 2020) to obtain informative gradients that lead to improved recommendations to target local interventions.

To summarize our contributions, in this paper we (1) formulate a training objective for differentiable models that recommend a top-K set of regions by directly optimizing the best-possible-reach (BPR) performance metric, (2) show how to use advances in perturbed optimizers to effectively do gradient-based learning on this objective, and (3) demonstrate how this framework can improve recommendations for targeted interventions in the real-world public health challenge of opioid-related overdose forecasting across 1620 census tracts in the state of Massachusetts.

## 2. Differentiable Top-K Optimization

We now review the essential background our approach builds upon from the growing field of differentiable top-K optimization (Cordonnier et al., 2021). The core task is a supervised learning task. For each instance, we are given an input $x$ representing observed measurements for $S$ locations (e.g. spatial regions of a state or country in a public health application). We first wish to build a scoring function $r_\theta(x) \in \mathbb{R}^S$ that scores each of the $S$ possibilities with a real number. Here, $\theta$ represents the learnable parameters of our scoring function (e.g. weights of a neural network). Then, using these scores, we wish to select a size-$K$ subset of all locations, denoted $\mathcal{R}$, to recommend for further action or processing. A natural way to do this selection is to keep the top $K$ highest scoring locations: $\mathcal{R}_\theta(x) = \text{TOPK}(r_\theta(x))$.

For training, we have available $T$ pairs of inputs $x_t$ and ground-truth outcomes $y_t$, plus a loss function $\ell$ that can grade each recommended set against outcome $y_t$. Model training then pursues minimizing the loss

$$\theta^* \leftarrow \operatorname*{argmin}_\theta \underbrace{\sum_{t=1}^T \ell(y_t, \mathcal{R}_\theta(x_t))}_{\mathcal{L}(\theta)}. \quad (1)$$

The key challenge to executing this training in practice is that the function TOPK is piece-wise constant, so gradients of $\mathcal{L}(\theta)$ with respect to $\theta$ are zero almost everywhere. Given a sub-optimal initial parameters $\theta$, trying to directly optimize Eq. (1) via gradient descent would lead nowhere

due to gradients evaluating to zero. To overcome this challenge, Cordonnier et al. (2021) leverage recent advances in perturbed optimizers (Berthet et al., 2020), also known as stochastic smoothing (Abernethy et al., 2016).

The first step is to reframe the top-K operation as a linear program (LP). Let vector $\mathbf{r} \in \mathbb{R}^S$ represent the real-valued scores for each location of interest. Let $\mathcal{C}_K \subset \mathbb{R}^S$ denote a constrained domain of "relaxed" indicator vectors:

$$\mathcal{C}_K = \{\boldsymbol{i} \in \mathbb{R}^S : \textstyle\sum_{s=1}^S i_s \leq K \text{ and } 0 \leq i_s \leq 1 \ \forall s\}$$

Then we can equivalently express the top-K operator on vector $\mathbf{r}$ as the solution to the following linear program (LP)

$$\text{LPTOPK}(\mathbf{r}) = \operatorname*{argmax}_{\boldsymbol{i} \in \mathcal{C}_K} \langle \boldsymbol{i}, \mathbf{r} \rangle, \quad \langle \boldsymbol{i}, \mathbf{r} \rangle \triangleq \textstyle\sum_{s=1}^S i_s r_s. \quad (2)$$

One of the possible equivalent solutions to Eq. (2) will yield a truly K-hot binary vector $\boldsymbol{i}$, where entries that are set to 1 indicate the $K$ chosen locations.

The next step is to apply perturbations. Let random variable $\mathbf{z} \in \mathbb{R}^S$ be Gaussian-distributed with zero mean and unit variance. We construct a perturbed linear program by averaging over $M$ iid samples of $\mathbf{z}$, denoted $\mathbf{z}^m \sim \mathcal{N}(0, I_S)$:

$$\text{PLPTOPK}(\mathbf{r}) = \frac{1}{M} \sum_{m=1}^M \operatorname*{argmax}_{\boldsymbol{i} \in \mathcal{C}_K} \langle \boldsymbol{i}, \mathbf{r} + \sigma \mathbf{z}^m \rangle, \quad (3)$$

where $\sigma > 0$ is a user-controlled hyperparameter setting the noise-level of the stochastic smoothing. This is an unbiased estimator of $\text{LPTOPK}(\mathbf{r})$. Like any $M$-sample Monte Carlo estimator, its variance around this ideal mean decreases as the number of samples $M$ increases, allowing the user a way to improve accuracy at the expense of runtime.

Putting it all together, when training during the forward pass we pursue a variant of Eq. (1) where we estimate the $K$-selected regions via $\mathcal{R}_\theta(x) = \text{PLPTOPK}(r_\theta(x))$. We set up this way because this perturbed definition yields the same selected regions as the ideal TOPK function in expectation, but crucially the perturbed operator is not piecewise-constant: gradients with respect to $\theta$ are no longer almost always zero. Following Cordonnier et al. (2021), we need not actually solve an LP for each of the $M$ perturbations; instead equivalently we can just compute TOPK directly on each perturbation.

For the backward pass of automatic differentiation, we need the Jacobian of the $S$-dimensional output vector $P$ produced by $\text{PLPTOPK}(\mathbf{r})$. Following Cordonnier et al. (2021), we have a straightforward expression

$$J(\mathbf{r}) = \nabla_\mathbf{r} P = \mathbb{E}_z \left[ \operatorname*{argmax}_{\boldsymbol{i} \in \mathcal{C}_K} \langle \boldsymbol{i}, \mathbf{r} + \sigma \mathbf{z} \rangle \mathbf{z}^\top / \sigma \right]. \quad (4)$$

Here, the Jacobian $J$ is an $S \times S$ matrix, where entry $j, k$ gives the scalar derivative $\frac{P_j}{r_k}$. This formula is derivable from Lemma 1.5 of Abernethy et al. (2016). Again, we compute an $M$-sample Monte Carlo estimate of this expectation in practice.

| | | |
| --- | --- | --- |
| 6.3 | 3.6 | 0.6 |
| 4.6 | 6.2 | 9.6 |
| 7.1 | 9.2 | 5.4 |

| | Mean Sq. Error | |
| Model | train | test |
| --- | --- | --- |
| MLP + Top-2 | 1.0 | 1.7 |
| MLP | 3.8 | 5.0 |

*Figure 1.* **Toy task: Add the 2 Most Yellow**. *Left:* Illustration of one instance $(x_t, y_t)$ for this task. Each feature vector $x_t$ consists of 9 numbers, each with a corresponding RGB color. The prediction goal is to return the sum of the 2 most-yellow values (circled). The outcome here is $y_t = 11.7$. *Right:* Predictive error (lower is better). Using a perturbed Top-2 operator yields better results.

**Demo on Toy Data: Add 2 Most Yellow Numbers.** In order to demonstrate differentiable top-K optimization on an open dataset, we now provide a synthetic prediction task where success requires explicit reasoning about top-K values. Inspired by the billiard ball experiment in Cordonnier et al. (2021) but wishing to avoid complexities of image processing, we generate a toy dataset of $(x_t, y_t)$ pairs as follows. First, for features $x_t$, we generate 9 numeric values uniformly in (0.0, 10.0). For each value, we also generate 3 intensities in (0.0, 1.0) representing red, green, and blue (R, G, B) color components. Each $x_t$ can thus be seen as a 36-dimensional feature vector, or artistically rendered as 9 colored numbers (Figure 1). Second, we define outcome $y_t \in \mathbb{R}$ as the sum of the two most-yellow numbers, where yellowness is defined as $R + G - B$. In order to generalize well to unseen data, model architectures can use a TOPK operator to indicate which numbers are the "2 most yellow".

To define the score function $r_\theta(x_t)$, we use an artificial neural network that takes only the 27 RGB values in $x_t$ and produces a score in (0.0, 1.0) for each of the 9 numbers. These scores are then element-wise multiplied with the 9 values, and the sum is taken. A perfect model would produce a score vector that is 2-hot, with a 1 at the location of the two most yellow values, and 0's elsewhere. We compare a 2-layer MLP that passes the score through the perturbed TopK module (with $K = 2$) to one without any inductive bias encouraging the score to be 2-hot.

Results of this demo are shown in Fig. 1 (right) after training on a dataset of $T = 5000$ instances. The addition of the top-2 module leads to lower MSE on test data, indicating our gradient-based learning can enable generalization.

## 3. Application to Opioid Overdose Forecasting

**Dataset.** We obtained death certificate data on fatal opioid-related overdoses for years 2001-2020 in Massachusetts. These overdose deaths were defined as unintentional, intentional, and undetermined drug poisonings containing an opioid code as an underlying cause-of-death. Such data are publicly available upon request from the MA Registry of Vital Records and Statistics. The data contains residential street addresses for each individual decedent, which we subsequently geocoded for spatial analyses. Our institution's IRB gave the project a Not Human Research Determination.

**Modeling Task.** We divided the state into the $S = 1620$ census tracts used in the 2020 U.S. Census. Each tract by design contains typically 4000 people (range 1200-8000) (US Census Bureau, 2022). We divided time into calendar years. We compute the observed number of death events $Y_{s,t}$ at time unit $t$ for individuals residing in census tract $s$, using open tools (Freeman, 2022) that call the US Census Geocoding API to map each residential street address to a census tract. Given counts observed over a training period up to time $t-1$, our goal is to predict death counts at time $t$ across all $S$ census tracts.

**Percentage of Best-Possible Reach.** The performance metric of interest is what Heuton et al. (2022) call the fraction or percentage of *best-possible reach*, hereafter abbreviated BPR. This is a way of judging in hindsight how well the model selects a targeted set of $K$ regions to intervene out of $S$ possible regions. The appropriate value of $K$ is decided in advance. Let $\mathcal{R}$ be a model's recommended set of $K$ distinct regions ($|\mathcal{R}| = K$). Let $\mathbf{Y}_t = [Y_{1,t}, \ldots Y_{S,t}]$ be the $S$-dimensional vector of adverse event counts (for our application, opioid-related overdose deaths) observed at the target time $t$ across all $S$ regions. Recall that the TOPK function returns the integer IDs of the $K$ largest entries in a given vector. We then define BPR as

$$\text{BPR}(\mathbf{Y}_t, \mathcal{R}) = \frac{\sum_{s \in \mathcal{R}} Y_{s,t}}{\sum_{s \in \text{TOPK}(\mathbf{Y})} Y_{s,t}}, \tag{5}$$

$$= \frac{\text{\# events in K regions picked by model}}{\text{\# events in actual K highest-count regions}}.$$

This fraction's numerator counts how many adverse events the model's recommendation could reach. The denominator counts how many adverse events a perfect oracle with knowledge of the future could reach on the same budget. For public health tasks, BPR has a convenient interpretation as the fraction of opioid-related overdoses the current model would identify compared to the best possible model. We often convert this fraction to a percentage, denoted %BPR. The best possible %BPR is 100.0, the worst possible is 0.0. An example calculation is shown in Appendix A.

Marshall et al. (2022) suggest a similar metric for their preregistered randomized controlled trial of opioid overdose forecasting in Rhode Island. The small difference is the denominator: they suggest summing over all $S$ regions, not just the top-K. We prefer the formulation in Eq. (5), because it defines a perfect %BPR always as 100%, and the range of valid BPR values does not change across time periods.

**Proposed models.** In order to directly predict where to intervene on this overdose application, we train neural net-

works by minimizing Eq. (1) via the methods in Sec. 2. For prediction at time $t$, we set as inputs $x_t$ the adverse event counts observed over the previous $W$ timesteps: $x_t = \boldsymbol{Y}_{1:S,t-W:t-1}$. This input is fed into a score function $r_\theta(x_t)$, whose top-K entries $\mathcal{R}(x_t)$ are compared to the actual outcome counts at the target time $y_t = \boldsymbol{Y}_{1:S,t}$ via loss $\ell(y_t, \mathcal{R}(x_t)) = 1 - \text{BPR}(y_t, \mathcal{R}(x_t))$. The two hyperparameters, the number of samples $M$ and smoothing noise $\sigma$ are set via grid search.

We consider two simple architectures for the score function $r$. First, a linear model over past mortality values: $r_s = \sum_{\tau=t-W}^{t-1} \theta_\tau Y_{s,\tau}$. Second, an MLP with 2 hidden layers (50 units then 10 units), whose weights $\theta$ are shared across all spatial locations: $r_s = \text{MLP}([\boldsymbol{Y}_{s,t-W:t-1}, u_s]; \theta)$. Here $u_s \in \mathbb{R}^5$ are additional covariates concatenated onto $\boldsymbol{Y}$, representing the 5-dimensional Social Vulnerability Index (CDC ATSDR, 2018) of tract $s$.

**Protocol.** The goal of our experiment is to predict future opioid overdose mortality based on historical data. For training, we assemble instances $(x_t, y_t)$ for 5 years (2013-2017), with historical covariates inside each $x_t$ available for up to $W = 5$ previous years. Hyperparameters are chosen using the $x_t, y_t$ pair from year 2018 for validation. Finally, models are evaluated on predictions the years 2019-2020. For each test year $t$, we predict using updated inputs $x_t$ but the same fixed parameters $\theta$ learned from training. In all experiments, we chose $K = 100$ via conversation with public health experts.

**Baselines.** We compare our proposed method to several baselines. First, we include a model that recommends $K$ tracts at random, to help gauge overall task difficulty. Second, we try risk scores determined by historical averages that need no training at all. Public health officials might reasonably suggest selecting the $K$ areas with the highest historical mortality. Thus, we include the historical 7-year average (7 chosen out of 1,3,5,7 by performance on the validation year).

We further include *probabilistic models* deliberately developed for forecasting count data. We fit a generalized linear model (GLM) and a gradient boosted tree (GBT), each maximizing Poisson likelihood. We also fit Heuton et al. (2022)'s Zero Inflated Gaussian Process (ZIGP) model developed specifically for opioid overdose forecasting, using code from the authors. (We use annual time units, so %BPR here is not comparable to Heuton et al.'s quarter-year analysis). These probabilistic baselines are fit using both past mortality and the Social Vulnerability Covariates.

**Results and Analysis.** Table 1 reports the %BPR averaged over both test years (2019, 2020). We first observe the challenging nature of fatal opioid overdose forecasting: simple historical averages are competitive with GLMs and *outper-*

| MODEL | OBJECTIVE | %BPR |
|---|---|---|
| RANDOM | N/A | 24.3 |
| 7-YEAR AVERAGE | N/A | 58.7 |
| MLP + TOPK | BPR | **63.4** |
| LINEAR + TOPK | BPR | 59.8 |
| GLM (LINEAR + POISSON) | POISSON LIK. | 60.2 |
| GRAD. BOOSTED TREES | POISSON LIK. | 56.4 |
| ZIGP | POISSON LIK. | 44.9 |

*Table 1.* Comparison of models in terms of percentage of best possible reach (%BPR, higher is better) averaged over test years 2019 and 2020, using $K = 100$ of $S=1620$ tracts in MA.

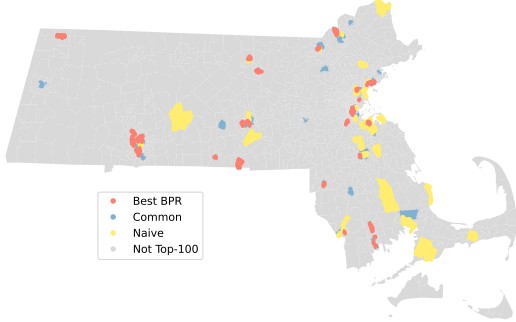

*Figure 2.* Visualizing tracts recommended by different models

*form* GBT ensembles of trees and the ZIGP. Yet we also see the strength of our proposed objective: the best performing model by a margin of 3 points is our 2-layer MLP trained directly to optimize BPR via perturbed top-K optimization.

To better evaluate the difference between our best-performing MLP trained to maximize BPR and the 7-year average model, a map of the different decisions is presented in Figure 2. The "naive" 7-year average model selects larger, rural tracts, responding to transient past spikes. Our "best BPR" model appears to focus on smaller urban tracts, capturing spatial patterns that yield better recommendations.

## 4. Conclusion

On one hand, it is not too surprising that a model trained to maximize BPR should have the highest BPR on heldout data. However, if metrics such as BPR are truly a useful evaluation criteria for public health interventions, as argued for in the large-scale randomized trial plan by Marshall et al. (2022), we argue that BPR as a metric should inform training, not just evaluation.

Prior to this paper, perturbed top-K techniques from the optimization community (Berthet et al., 2020; Abernethy et al., 2016; Cuturi et al., 2019) have not to our knowledge been applied to spatiotemporal problems in public health. We hope these methods can lead to improved decision-making about where to intervene, as data-driven mitigation strategies are desperately needed to reduce the harms of the opioid epidemic.

## Acknowledgements

Author KH is supported by the U.S. National Science Foundation under NSF award NRT-HDR 2021874. Authors KH, TJS, and MCH gratefully acknowledge support for early work on this project from NSF award IIS-1908617.

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

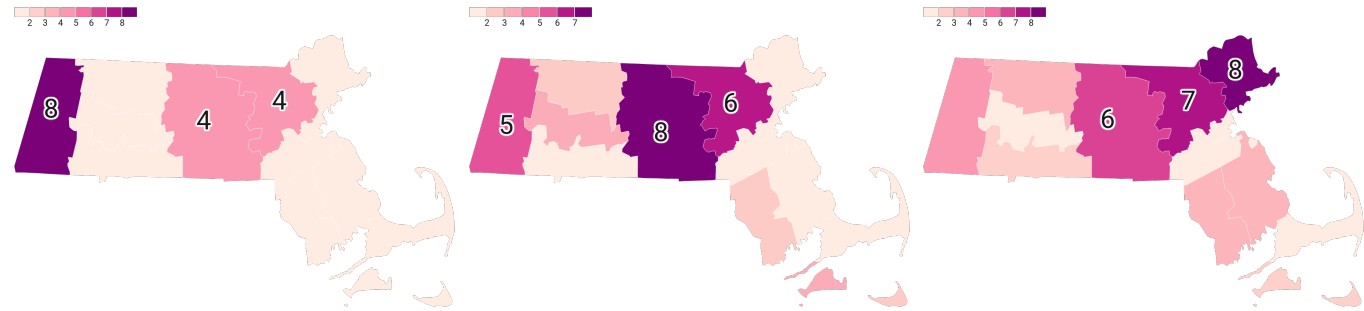

*Figure 3.* A cartoon illustration of a BPR calculation for Massachusetts counties. Here we demonstrate the BPR calculation for $K = 3$. *Left*: a synthetic ground-truth dataset. The top-3 true counties are labeled, and all other values are 1. The denominator of Eq. 5 will be $8 + 4 + 4 = 16$.
*Center*: A hypothetical forecast, attempting to predict the left map. Although the magnitude of every prediction is wrong, this forecast correctly identifies the top 3 counties and would achieve a Perfect BPR of %100.
*Right*: Another hypothetical forecast, but this time only two of the top-3 counties are correctly identified. To calculate the numerator of Eq. 5, we sum the ground-truth counts from the predicted top-3 locations: $4 + 4 + 1 = 9$, resulting in a BPR of $\frac{9}{16} = \%56.25$

## A. Example BPR Calculation

Figure 3 shows a hypothetical scenario to demonstrate the calculation of BPR for the top-3 locations. The leftmost map depicts the synthetic "ground-truth." The top-3 counties are labeled. When calculating BPR, we always use the counts from the ground truth.

The denominator of BPR is the sum of the top-K locations in the ground truth data:

$$\text{BPR}(\mathbf{Y}_t, \mathcal{R}) = \frac{\sum_{s \in \mathcal{R}} Y_{s,t}}{\sum_{s \in \text{TOPK}(\mathbf{Y})} Y_{s,t}} = \frac{\sum_{s \in \mathcal{R}} Y_{s,t}}{8 + 4 + 4}$$

Given the Figure 3 center forecast, the recommended top-3 locations $\mathcal{R}$ are exactly the same as the TOP3$(\mathbf{Y})$ locations, so the BPR is perfect:

$$= \frac{8 + 4 + 4}{8 + 4 + 4} = \%100$$

Given the Figure 3 right forecast, the recommended top-3 locations $\mathcal{R}$ only contain two of the true TOP3$(\mathbf{Y})$ locations. Again, to calculate the numerator, we take the ground-truth counts from the recommended locations, but this time they fall short of a perfect BPR:

$$= \frac{4 + 4 + 1}{8 + 4 + 4} = \frac{9}{16} = \%56.25$$

## B. Selection of $\sigma$ and $M$ Hyperparameters

The two most important hyperparameters for the perturbed Top-K optimizer PLPTOPK are the noise level $\sigma$ and number of samples $M$. For the opioid data experiment a hyperparameter grid of $\sigma = [0.05, 0.1, 0.3, 0.5]$ and $M = [25, 50, 100, 500]$ was used. For both models which used the perturbed Top-K optimizer, $\sigma = 0.3$ and $M = 50$ was selected.

