# OpenReview forum: "Learning where to intervene with a differentiable top-k operator: Towards data-driven strategies to prevent fatal opioid overdoses"
_ICML.cc/2023/Workshop/IMLH — IMLH 2023 PosterShortPaper_

### Official Review · Reviewer_gUWG · 2023-06-11
**Intuitive and Effective Methodologies, Potentially Impactful Application.**

**Rating:** 7
**Confidence:** 4

**Review:**

This short 4-page paper aims to reduce the cost of interventions carried out by public health organizations in different regions. Building on the metrics proposed in recent research, the authors utilize machine learning techniques, including a differentiable top-K operator, to optimize these metrics.

Strengths:

•	The format of this paper is well-organized and the writing is very clear. The image in the paper is visually appealing and engaging.

•	The paper provides a thorough exposition of the methodology, with well-explained formulas.

•	The explanation of the experiments and results in the article is comprehensive.

•	The paper presents a novel idea that can potentially have a positive impact on cost reduction in the industry, making it a valuable contribution.

Weaknesses/Suggestions:

•	There is no major flaw in the simple and intuitive experiments provided. Though, Table 1 suggests that higher expressivity of the model would benefit BPR. Therefore, the authors are encouraged to explore the limits of the model they can use, like deep nets, for the best BPR they can reach.

---

### Official Review · Reviewer_zk2c · 2023-06-15
**A great application of differentiable top-k optimization: opioid overdose forecasting**

**Rating:** 6
**Confidence:** 3

**Review:**

This paper proposes a method for learning from historical opioid overdose events to predict where to intervene. The method is built upon differentiable top-k optimization.

The paper is well-written and easy to follow. The empirical results on the real-world data are strong, showing it a promising direction to apply differentiable top-k optimization into the task of opioid overdoes forecasting.

---

### Official Review · Reviewer_hWiq · 2023-06-17
**Towards data-driven strategies to prevent fatal opioid overdoses**

**Rating:** 6
**Confidence:** 3

**Review:**

(+) This paper's focus may be interesting and motivate further research into the underlying problem.

(+) The paper's writing level is satisfactory.

(+) The demo on Toy Data seems intuitive.

(-) The introduction does not clearly convey the researcher's intuition or goals. I would recommend that the authors revise the paper's abstract, introduction, and conclusion to provide a clearer understanding of the paper's objectives and main messages. Additionally, a more insightful analysis and consideration of various public datasets to investigate the generalizability of the paper's findings would significantly enhance its credibility.

---

### Meta-Review · Program_Chairs · 2023-06-19

**Recommendation:** Accept (Poster)
**Confidence:** 4

**Metareview:**

This paper aligns well with the interest of this workshop. Moreover, this work receives consistent positive reviews. Thus I recommend acceptance.

---

### Decision · Program_Chairs · 2023-06-20

Accept (Poster Short Paper)